

# Revegetation pattern affecting accumulation of organic carbon and total nitrogen in reclaimed mine soils

Ping Ping Zhang[1,2,3], Yan Le Zhang[3,4], Jun Chao Jia[3,4], Yong Xing Cui[3,4], Xia Wang[3,4], Xing Chang Zhang[3,4] and Yun Qiang Wang[1,2]

[1] State Key Laboratory of Loess and Quaternary Geology, Institute of Earth Environment, Chinese Academy of Sciences, Xi'an, China

[2] CAS Center for Excellence in Quaternary Science and Global Change, Xi'an, China

[3] Institute of Soil and Water Conservation, Chinese Academy of Science and Ministry of Water Resources, Yangling, China

[4] University of Chinese Academy of Sciences, Beijing, China

Corresponding author
Yun Qiang Wang, wangyq@ieecas.cn, wangyunq04@163.com

## ABSTRACT

Selecting optimal revegetation patterns, i.e., patterns that are more effective for soil organic carbon (SOC) and total nitrogen (TN) accumulation, is particularly important for mine land reclamation. However, there have been few evaluations of the effects of different revegetation patterns on the SOC and TN in reclaimed mine soils on the Loess Plateau, China. In this study, the SOC and TN stocks were investigated at reclaimed mine sites (RMSs), including artificially revegetated sites (ARSs) (arbors (Ar), bushes (Bu), arbor-bush mixtures (AB), and grasslands (Gr)) and a natural recovery site (NRS), as well as at undisturbed native sites (UNSs). Overall, the SOC and TN stocks in the RMSs were lower than those in the UNSs over 10–13 years after reclamation. The SOC stocks in the RMSs and UNSs only differed in the top 0–20 cm of the soil ($p < 0.05$). Except for those in Ar, the SOC and TN stocks in the ARSs were significantly larger than those in the NRS ($p < 0.05$). Compared with those in the NRS, the total SOC stocks in the 100 cm soil interval increased by 51.4%, 59.9%, and 109.9% for Bu, AB, and Gr, respectively, and the TN stocks increased by 33.1%, 35.1%, and 57.9%. The SOC stocks in the 0–100 cm soil interval decreased in the order of Gr (3.78 kg m$^{-2}$) > AB (2.88 kg m$^{-2}$) ≥ Bu (2.72 kg m$^{-2}$), and the TN stocks exhibited a similar trend. These results suggest that grasslands were more favorable than woodlands for SOC and TN accumulation in this arid area. Thus, in terms of the accumulation of SOC and TN, grassland planting is recommended as a revegetation pattern for areas with reclaimed mine soils.

## INTRODUCTION

With growing global warming, soil organic carbon (SOC) has received increasing attention worldwide. As the largest terrestrial pool of organic carbon, a slight change in SOC pool could significantly alter the atmospheric $CO_2$ concentration and exert great effect on global climate (*Lal, 2004*). Besides, SOC and total nitrogen (TN) are key indicators of the fertility and quality of soil, and they are highly important for improving soil structure and

increasing crop productivity (*Jia et al., 2012*). Therefore, understanding the SOC and TN concentration and stock in ecosystems is essential for alleviating the global climate change and determining the sustainable management of land resources (*Zhang et al., 2018a*; *Zhang et al., 2018b*).

The adverse effects of land degradation on SOC and TN cycles have received considerable scientific attention worldwide in recent decades. Coal mining, especially surface mining, can considerably alter habitat and cause land degradation (*Mukhopadhyay & Masto, 2016*; *Liu et al., 2017*). Mining eliminates vegetation, disrupts soil profiles, and changes topography and geological permanently (*Bao et al., 2017*; *Ahirwal & Maiti, 2018*). These processes further enhance soil erosion phenomena, interrupt the natural carbon (C) and nitogen (N) cycles, and reduce the SOC and TN pools (*Shrestha & Lal, 2010*). However, appropriate reclamation approaches and management practices can reduce the soil degradation and increase the accumulation of SOC and TN (*Ahirwal, Maiti & Reddy, 2017*; *Ahirwal, Maiti & Singh, 2017*).

Unlike naturally formed soils, reclaimed mine soils are pedogenically young soils, which are highly disturbed and artificially constructed from the materials excavated during mining (*Shrestha & Lal, 2008*; *Cao et al., 2015*). They are often characterized by poor chemical, physical, and biological properties, such as low nutrient levels, high toxic elements contents, poor soil structure and aggregation, increased soil compaction, low water-holding capacity, and reduced microbial activity (*Ahirwal & Maiti, 2016*; *Zhou et al., 2017*; *Wang et al., 2018*). These poor soil properties can adversely affect the growth of plants as well as delaying the development of the soil profile and the accumulation of SOC and TN through natural succession (*Ahirwal & Maiti, 2018*). However, anthropogenic intervention such as artificial revegetation can accelerate the restoration process (*Jia et al., 2012*; *Mukhopadhyay & Masto, 2016*).

The effects of mining (*Shrestha & Lal, 2011*) and reclamation followed by the establishment of forests (*Ussiri, Lal & Jacinthe, 2006*; *Mukhopadhyay & Masto, 2016*; *Ahirwal, Maiti & Reddy, 2017*; *Ahirwal, Maiti & Singh, 2017*; *Ahirwal & Maiti, 2018*), grasslands (*Evanylo et al., 2005*; *Yang et al., 2015*), and croplands (*Yuan et al., 2017*) on the SOC and TN pools have been assessed in many studies. However, the soils planted under various revegetation patterns have different SOC and TN stocks and sequestration capacities due to their different erosion rates, deposition rates, and the input quantities of above- and belowground litter (*Wei et al., 2009*; *Maraseni & Pandey, 2014*). Thus, selecting optimal revegetation patterns (patterns that are more effective for SOC and TN accumulation) is particularly important for reclamation (*Ahirwal, Maiti & Reddy, 2017*). The optimal revegetation patterns have been shown to be site specific, because of the different nature of the spoil material and geo-climatic condition (*Shrestha & Lal, 2007*; *Chatterjee et al., 2009*; *Ganjegunte et al., 2009*; *Shrestha & Lal, 2010*; *Datta et al., 2015*). These differences strongly hinder the application of specific successful reclamation patterns in other areas.

The northwestern Loess Plateau in China is a transition zone between arid and semiarid regions with a fragile environment. During recent decades, land degradation and soil erosion have increased in this region due to the expansion of coal mining. The Chinese government established a series of laws and regulations, such as the ''Land Reclamation Regulations''

in 1988 (*He et al., 1996*), in order to improve the regional ecological environment, alleviate the land degradation, and promote the sequestration of SOC and TN. Subsequently, the reclamation and revegetation of severely disturbed mining areas has attracted increased attention, and measures have been implemented for artificial revegetation in this region, such as the planting of arbors, bushes, and grasses, and natural vegetation recovery. The effects of revegetation and time on the properties of reclaimed mine soils have been studied widely (*Zhao et al., 2013*; *Wang, Jiao & Bai, 2014*; *Huang et al., 2016*; *Liu et al., 2017*; *Yuan et al., 2017*). However, there have been few evaluations of the effects of different revegetation patterns on the SOC and TN pools in reclaimed mine soils.

Therefore, in the present study, we determined the SOC and TN concentrations and stocks in reclaimed mine sites (RMSs) and in adjacent undisturbed native sites (UNSs) at the Heidaigou surface coal mine. The construction of this mine began in 1990 and it is the largest surface coal mine in China, with an annual output of 25 million tons (*Li et al., 2014*). The objectives of this study were: (1) to evaluate the effects of different revegetation patterns on the SOC and TN distributions and stocks in RMS soils; and (2) to assess the changes in the SOC and TN stocks in RMS soils relative to those in UNS soils.

## MATERIALS & METHODS

### Study area

The study was conducted in Heidaigou surface mine within the Junggar Banner in Erdos on the Loess Plateau, China (39°43′–39°49′N and 111°13′–111°20′E; Fig. 1), and was approved by the land environmental protection office of Shenhua Group Zhungeer Energy Co. (project number: KZCX2-XB3-13-02). The mine encompasses a total area of 52.11 km$^2$, and the mine elevation is between 1025 m and 1302 m above sea level. The site has a temperate continental arid climate with a mean annual temperature of 7.2 °C. The mean annual precipitation is 404.1 mm (ranging from 213.0 to 459.5 mm), with 60–70% received during the growing season between July and September. The average annual evaporation is 1943.6 mm and the relative humidity is 58%. The wind is mostly calm, blowing in a north–northwest direction at an average speed of 2.2 m s$^{-1}$. The predominant soil series prior to mining was mainly loess soil (Calcaric Regosol, *FAO/UNESCO, 1988*).

The mine has six overburden dumps, and the northern dump was selected for this study (Fig. 1). Since 1993, large-scale reclamation and revegetation activities have been implemented at this dump for the purpose of soil ripening, and by 2005, the reclamation area had reached 1.48 km$^2$. In the process of reclamation, at least 1 m of top subsoil (the deep loess parent materials stripped during mining) was applied on the top of the dump and appropriate leveling was conducted to create a flat surface. Four predominant artificial revegetation patterns were selected and compared to the natural recovery pattern to evaluate their effects on SOC and TN accumulation: the planting of arbors (Ar), bushes (Bu), arbor–bush mixtures (AB), and grasslands (Gr). This study was not a replicated field plot experiment or an established design. Therefore, the vegetation types comprising Ar, Bu, AB, and Gr were established at four, four, two, and three typical vegetation collocations, respectively, as pseudo-replications due to the lack of true replicates, with a total of 13

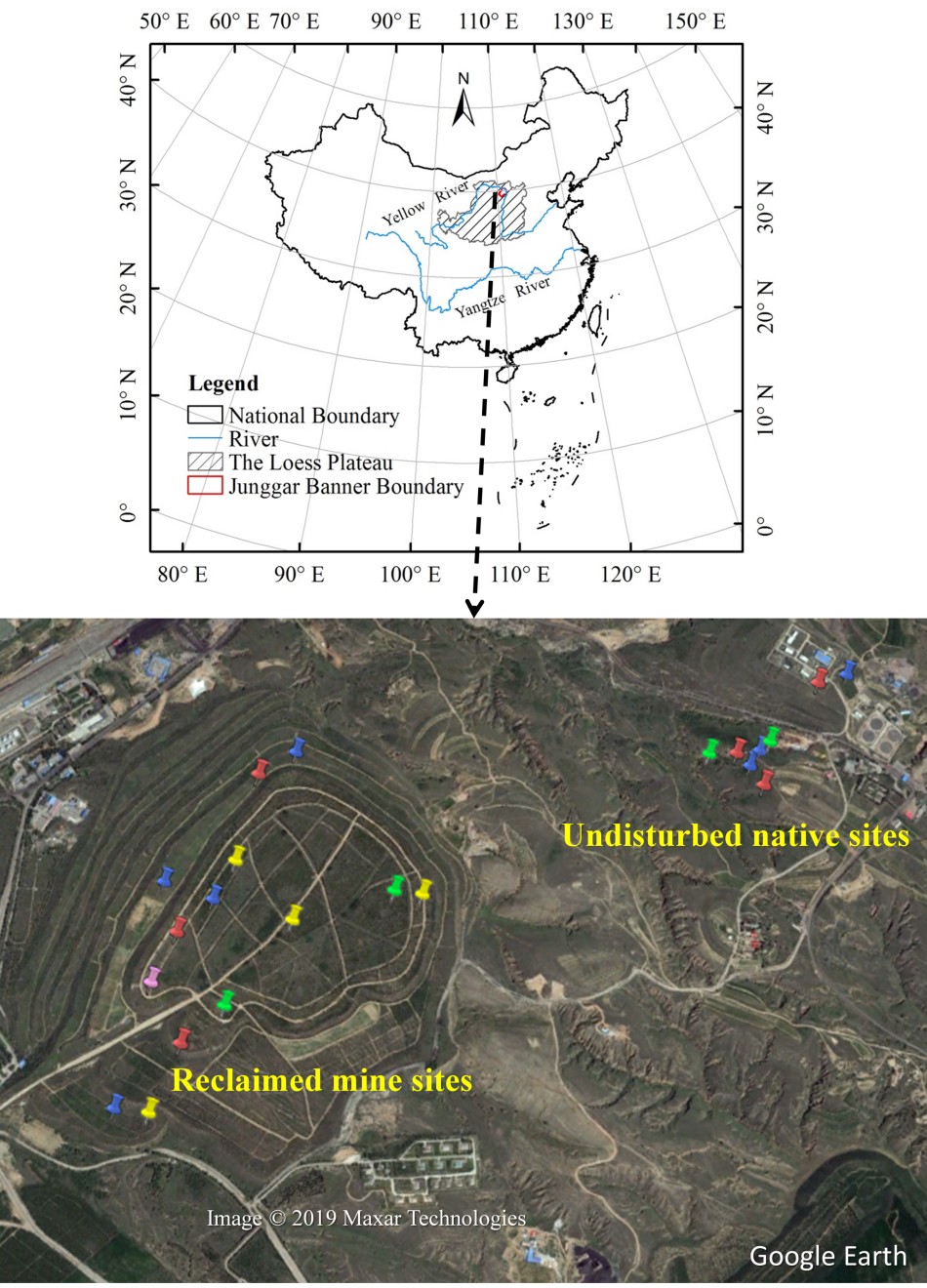

**Figure 1** **Locations of sampling sites at the Heidaigou surface mine on the Loess Plateau, China.** The yellow, blue, green, red, and pink symbols represent the sites planted with trees, bushes, arbor–bush mixtures, grasses, and the natural recovery site, respectively. Map credit: ©2019 Maxar Technologies and Google Earth.

artificially revegetated sites (ARSs). Revegetation at these selected ARSs was implemented between 2002 and 2005. The vegetation was sparse and it only included scattered *Leymus chinensis* (Trin.) Tzvel. and *Leymus secalinus* (Georgi) Tzvel., so only one experimental

site was established for the natural recovery approach and it was designated as the natural recovery site (NRS). The NRS was not planted and it had been unmanaged since 2004. All of the selected RMSs had similar soils and terrain conditions, which provided an excellent opportunity to compare the different revegetation patterns. The UNSs located within 2 km of the reclaimed dump and without significant disturbance due to mining activities were included in the study as reference sites. Historically, the UNSs were mainly sloping farmlands, which were later converted into forests or grasslands due to the implementation of the state-funded "Grain-for-Green project" in 1999. Three Bu sites, two AB sites, and three Gr sites with vegetation type changes since approximately 2001 were selected for this study. In total, 22 experimental sites were established in this study (Fig. 1). Global Positioning System coordinates (with a resolution of 3 m) were recorded for each sampling location, and the details of these sites are shown in Table 1.

## Soil sampling and analysis

At each RMS and UNS, disturbed soil samples were collected at depths of 0–10, 10–20, 20–40, 40–60, and 60–100 cm from five randomly selected locations using a 20 cm by 5 cm soil auger in July 2015. The selected replicate locations were at least 20 m from the boundary of the site (from the area where the vegetation type changed) and they were separated by a distance of approximately 10 m to account for spatial variability. Soil samples were collected from Ar, Bu, and AB sites near the center of the inter-tree space. In total, 550 soil samples were collected. The soil samples were air-dried, crushed, and passed through 0.25–mm sieves before performing SOC and TN measurements. The SOC concentration (g kg$^{-1}$) was measured using the Walkley–Black method (*Nelson & Sommers, 1982*), and the TN concentration (g kg$^{-1}$) was determined using the semi-macro Kjeldahl method (*Bremner & Mulvaney, 1982*).

One 1 m deep pit was dug at one typical site of each vegetation type in the RMS and UNS, and core samples of undisturbed soil (100 cm$^3$) were collected at depths of 0–10, 10–20, 20–40, 40–60, and 60–100 cm. Three replicate measurements were conducted in each layer. The soils were transported to the laboratory and dried to a constant weight at 105 °C to calculate the bulk density (g cm$^{-3}$). The SOC stock (SOC$_i$) and TN stock (TN$_i$) (kg m$^{-2}$) were then calculated using the following formula:

$$SOC_s = \sum \frac{SOC_i \times BD_i \times d_i}{100}$$

$$TN_s = \sum \frac{TN_i \times BD_i \times d_i}{100}$$

where SOC$_i$, TN$_i$, BD$_i$, and d$_i$ represent the SOC concentration (g kg$^{-1}$), TN concentration (g kg$^{-1}$), bulk density (g cm$^{-3}$), and thickness (cm) of the $i$ th layer, respectively.

At each RMS and UNS, aboveground leaf litter was also collected from five randomly selected quadrants measuring 1 m × 1 m. The leaf litter was placed in mesh bags and transported to the laboratory to determine the dry mass (g m$^{-2}$) by weighing the litter after oven drying at 60 °C for 48 h.

**Table 1** Details of the vegetation collocations for the different vegetation types at reclaimed mine sites (RMSs) and undisturbed native sites (UNSs).

| Vegetation types | Vegetation collocations |
|---|---|
| | RMSs |
| Ar | *Populus alba* var. pyramdalis (H:10–12 m, RS:2.5 m, LS:3.5 m), planted in 2005 |
| | *Pinus tabuliformis* Carr. (H: 3.0–3.5 m, RS: 2.0 m, LS: 2.5 m), planted in 2004 |
| | *Populus alba* var. pyramdalis (H: 10–12 m, RS: 3.0 m, LS: 3.0 m) interplanting with *Pinus tabuliformis* Carr. (H: 1.5–2.5 m, RS: 3.0 m, LS: 3.0 m), planted in 2004 |
| | *Robinia pseudoacacia* Linn. (H: 3.0–5.0 m, irregularly distributed, D: 25 trees/100 m$^2$), planted in 2004 |
| Bu | *Rhus typhina* L. (H: 2.5–3.0 m, RS: 3.0 m, LS: 3.0 m), planted in 2005 |
| | *Sabina vulgaris* Ant. (H: 0.6–0.9 m, RS: 2.0 m, LS: 2.0 m), planted in 2004 |
| | *Armeniaca sibirica* (L.) Lam. (H: 4.0–4.5 m, RS: 2.0 m, LS: 1.5 m), planted in 2004 |
| | *Armeniaca sibirica* (L.) Lam. (H: 3.0–5.0 m, RS: 2.5 m, LS: 2.5 m) interplanting with *Hippophae rhamnoides* Linn. (H: 1.5–2.0 m, RS: 2.5 m, LS: 2.5 m), planted in 2002 |
| AB | *Pinus tabuliformis* Carr. (H: 2.5–3.5 m, RS: 2.0 m, LS: 2.0 m) interplanting with *Armeniaca sibirica* (L.) Lam. (H: 2.0–3.0 m, RS: 2.0 m, LS: 2.0 m), planted in 2004 |
| | *Populus alba* var. pyramdalis (H: 8.0 m, RS: 2.5 m, LS: 3.0 m) + *Hippophae rhamnoides* Linn. (H: 2.0 m, irregularly distributed, 10 trees/ 100 m$^2$), planted in 2004 |
| Gr | Dominated by *Leymus chinensis* (Trin.) Tzvel. + *Medicago sativa* L. (coverage = ca 95%), planted in 2004 |
| | Dominated by *Leymus chinensis* (Trin.) Tzvel. + *Leymus secalinus* (Georgi) Tzvel. + *Artemisia argyi* H. Lév. & Vaniot (coverage = ca 95%), planted in 2004 |
| | Dominated *by Artemisia argyi* H. Lév. & Vaniot (coverage = ca 95%), planted in 2004 |
| NRS | This site was not planted and it was unmanged since 2005 (coverage = ca 25%) |
| | UNSs |
| Bu | *Armeniaca sibirica* (L.) Lam. (H:2.0–2.5 m, RS:1.2 m, LS:1.2 m), planted in 2001 |
| | *Caragana korshinskii* Kom. (H: 1.2–2.5 m, irregularly distributed, D: 15 trees/100 m$^2$), planted in 2001 |
| | *Hippophae rhamnoides* Linn. (H: 1.2–2.5 m, irregularly distributed, D: 28 trees/100 m$^2$), planted in 2001 |
| AB | *Pinus tabuliformis* Carr (H: 5.0-6.0 m, D: 15 trees/100 m$^2$) + *Armeniaca sibirica* (L.) Lam. (H: 3.0–5.0 m, D: 22 trees/100 m$^2$), planted in 2001 |
| | *Populus alba* var. pyramdalis (H: 4.8–5.8 m, D: 15 trees/100 m$^2$) + *Armeniaca sibirica* (L.) Lam (H: 3.2–4.5 m, D: 12 trees/100 m$^2$), planted in 2001 |
| Gr | Dominated by *Leymus secalinus* (Georgi) Tzvel. + *Melilotus officinalis* L. (coverage = ca 90%), planted in 2001 |
| | Dominated by *Stipa bungeana* Trin. (coverage = ca 95%), planted in 2001 |
| | Dominated by *Agropyron mongolicum* Keng (coverage = ca 95%), planted in 2001 |

**Notes.**

Ar, arbors; Bu, bushes; AB, arbor-bush mixtures; Gr, grasslands; NRS, natural recovery site; H, height; RS, row spacing; LS, line spacing; D, density.

## Statistical analysis

All data were tested for normality and homogeneity of variance. One-way analysis of variance (ANOVA) and least significant difference (LSD) tests ($p < 0.05$) were used to detect differences in the concentrations and stocks of SOC and TN associated with different revegetation patterns and soil depths. Few typical vegetation collocations were available for each land and the NRS was not field replicated, so the five sampling locations within each site were also used as pseudo-replicates for the statistical analysis. The independent sample $t$-test was applied to compare the concentrations and stocks of SOC and TN between RMSs

and UNSs, and only the soil values obtained at the Bu, AB, and Gr sites were used. All statistical analyses were performed using SPSS 16.0 software.

## RESULTS

### SOC and TN concentrations under different revegetation patterns

The concentrations of SOC and TN in the soil profile are shown in Table 2. The SOC concentration was strongly stratified based on soil depth for all of the revegetation patterns. The SOC concentrations were highest in the top depth (0–10 cm) and decreased as the soil depth increased, although a slight increase was observed at a depth of 60–100 cm for Bu and AB. Significant decreases occurred in the 0–40 cm depth for Gr and the 0–20 cm depth for the other four revegetation patterns. The SOC concentration did not differ significantly below 40 cm or 20 cm. As shown in Table 2, the revegetation pattern had a significant effect on the SOC concentration ($p < 0.05$). At a depth of 0–40 cm, the average SOC concentration ranged from 3.54 g kg$^{-1}$ for Gr to 1.50 g kg$^{-1}$ for NRS and it was ranked in the following order: Gr > Bu > AB > Ar > NRS. The average SOC concentrations (0–40 cm) for Gr, Bu, AB and Ar were approximately 136.5%, 74.4%, 55.9%, and 21.5% greater, respectively, than that for the NRS. Except for AB at depths of 10–20 and 20–40 cm and Ar, the SOC concentrations in the ARSs were significantly higher than those in the NRS. Among the ARSs, Gr contained significantly higher SOC concentrations than all the other revegetation patterns at all depths. The SOC concentration in Ar was significantly lower than that in Bu at all depths but lower than that in AB at a depth of 0–10 cm. No significant differences were observed between Bu and AB at any depth. Below 40 cm, the differences in the SOC concentration were smaller among the revegetation patterns. In addition, no significant differences were found between the ARSs and the NRS, except for Gr. Gr had the highest SOC concentration among the revegetation sites, it did not differ significantly from those in AB and Bu. In addition, no differences were observed between Ar, Bu, and AB.

The soil TN concentration also varied significantly according to the soil depth and revegetation pattern ($p < 0.05$, Table 2). The TN concentration exhibited a significant decreasing trend from 0–40 cm for Ar, Bu, and Gr and from 0–20 cm for AB and the NRS. The soil TN concentration in the 0–40 cm interval exhibited a similar trend to that of SOC and it was ranked in the following order: Gr > Bu > AB > Ar > NRS. The TN concentrations in the ARSs with Bu and Gr at all depths and with AB at a depth of 0–10 cm were significantly greater than those in the NRS, which was consistent with the SOC results. However, unlike SOC, the highest soil TN concentration in Gr was observed in the 0–20 cm depth. The differences among the revegetation patterns decreased below 40 cm and no differences were observed below 60 cm.

### SOC and TN stocks under different revegetation patterns

The average SOC stock varied from 0.52–1.17 kg m$^{-2}$ in the top 0–20 cm of the soil and from 1.80 to 3.78 kg m$^{-2}$ in the entire soil profile (0–100 cm) with the different revegetation patterns (Fig. 2). The SOC stock in the 0–20 cm depth accounted for 26.4–32.8% of the total SOC stock in the 0–100 cm depth. Except for those in Ar, the SOC stocks in the ARSs

**Table 2** Soil organic carbon (SOC) and total nitrogen (TN) concentration (g kg$^{-1}$) at different soil depths with different revegetation patterns.

| | Soil depth (cm) | Ar | Bu | AB | Gr | NRS |
|---|---|---|---|---|---|---|
| SOC | 0–10 | 2.71cA | 3.68bA | 3.54bA | 4.67aA | 1.90cA |
| | 10–20 | 1.52cB | 2.31bB | 1.84bcB | 3.44aB | 1.45cB |
| | 20–40 | 1.22cB | 1.84bBC | 1.62bcB | 2.51aC | 1.13cB |
| | 40–60 | 1.14bB | 1.34bC | 1.53abB | 1.88aC | 1.14bB |
| | 60–100 | 1.14bB | 1.46abC | 1.77aB | 1.79aC | 1.10bB |
| TN | 0–10 | 0.29bcA | 0.35bA | 0.34bA | 0.44aA | 0.21cA |
| | 10–20 | 0.19cB | 0.25bB | 0.21bcB | 0.31aB | 0.16cB |
| | 20–40 | 0.15bC | 0.19aC | 0.18abB | 0.20aC | 0.14bB |
| | 40–60 | 0.14bC | 0.16abC | 0.18aB | 0.18aC | 0.13bB |
| | 60–100 | 0.15aC | 0.16aC | 0.17aB | 0.16aC | 0.14aB |

**Notes.**

Ar, arbors; Bu, bushes; AB, arbor-bush mixtures; Gr, grasslands; NRS, natural recovery site.

Values followed by the same lower-case letters in rows and upper-case letters in columns do not differ significantly at $p < 0.05$.

were significantly greater than those in the NRS in both the 0–20 cm and 0–100 cm depths. The total SOC stocks at 100 cm with Bu, AB, and Gr were 51.4%, 59.9%, and 109.9% higher, respectively, compared with that in the NRS. Gr had the highest SOC stocks, which were 38.7% and 31.3% higher than those for Bu and AB, respectively, at a depth of 0–100 cm. However, the SOC stocks in Bu and AB were not significantly different.

The average soil TN stocks varied between 0.06 and 0.11 kg m$^{-2}$ in the 0–20 cm interval and between 0.22 and 0.34 kg m$^{-2}$ in the 0–100 cm interval (Fig. 2). The TN stock in the 0–20 cm depth accounted for 26.5–31.7% of the total TN stock in the 0–100 cm depth. Similar to the SOC stocks, the TN stocks were higher in the ARSs than those in the NRS, excluding those in Ar. Gr had significantly higher TN stocks compared with Bu and AB, but the TN stocks did not differ between Bu and AB.

## Comparison of SOC and TN concentrations and stocks in RMSs and UNSs

As shown in Figs. 3 and 4, there were significant differences in the SOC and TN concentrations and stocks between the RMSs and UNSs ($p < 0.05$). The average SOC concentration in the 0–100 cm interval was lower in the RMSs than that in the UNSs , i.e., 24.8% lower for Bu, 30.5% lower for AB, and 31.0% lower for Gr. However, the differences in the SOC concentrations between the RMSs and UNSs varied among the vegetation types and soil depths, but a significant difference was only observed for AB and Gr, and it was mainly in the 0–20 cm soil interval. For 0–20 cm, the RMSs contained significantly lower SOC stocks than the UNSs for AB and Gr, but the SOC stocks were similar to those in the UNSs for Bu. However, in the 0–100 cm interval, there were no significant differences in the SOC stocks between the RMSs and UNSs for all vegetation types.

The mean TN concentration in the 0–100 cm soil interval was lower in the RMSs than the UNSs, i.e., 29.4% lower for Bu, 27.8% lower for AB, and 43.1% lower for Gr (Fig. 3). In particular, the RMSs had significantly lower TN concentrations than the UNSs up to a

(A)

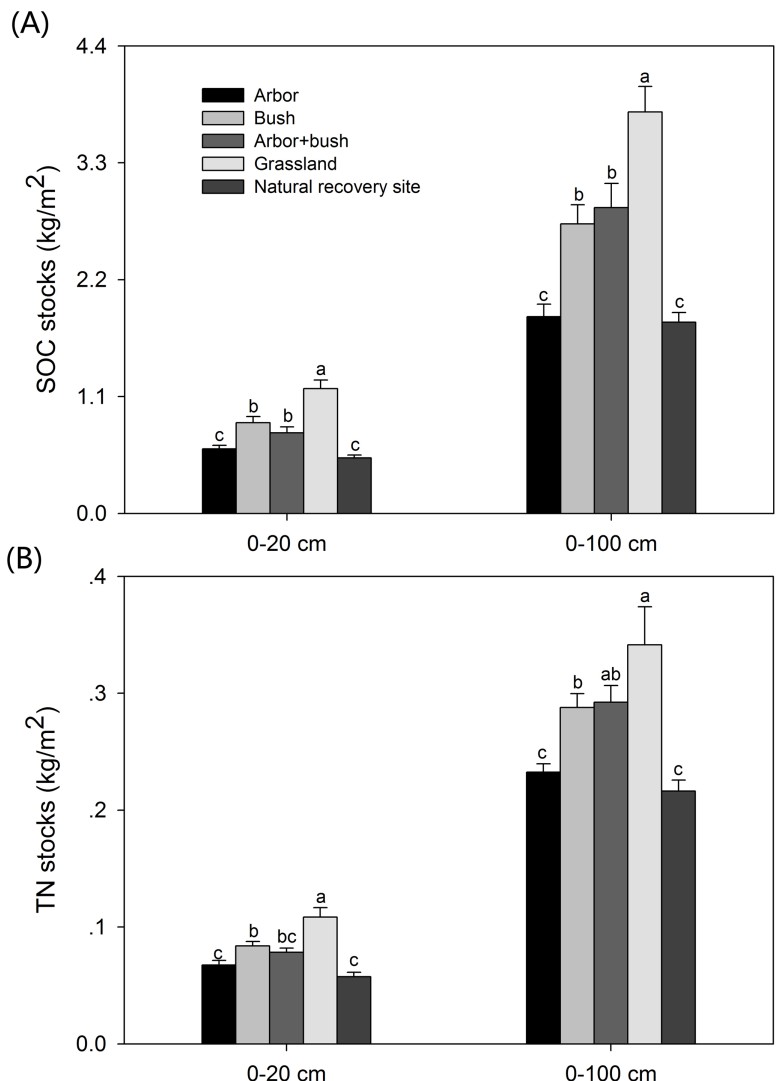

(B)

**Figure 2** **(A) Soil organic carbon (SOC) and (B) total nitrogen (TN) stocks (kg m$^{-2}$) at 0–20 and 0–100 cm soil depths under different revegetation patterns.** Different lowercase letters denote significant differences between revegetation patterns ($p < 0.05$). Error bars show the standard deviation.

depth of 100 cm for Bu and Gr and up to a depth of 40 cm for AB. In contrast to the SOC, the TN stocks in the RMSs were significantly lower than those in the UNSs in both the 0–20 cm and 0–100 cm intervals (Fig. 4).

## Comparison of aboveground plant litter in RMSs and UNSs

There were significant differences in the amounts of aboveground plant litter between the RMSs and UNSs ($p < 0.05$). The amounts in the ARSs were significantly higher than those in the UNSs for most cases, whereas the amounts in the NRS were significantly lower than those in the UNSs. Among the ARSs, the woodlands, i.e., A, B, and AB, had significantly higher amounts of aboveground plant litter than the grassland, i.e., G. However, no significant differences were found among the A, B, and AB.

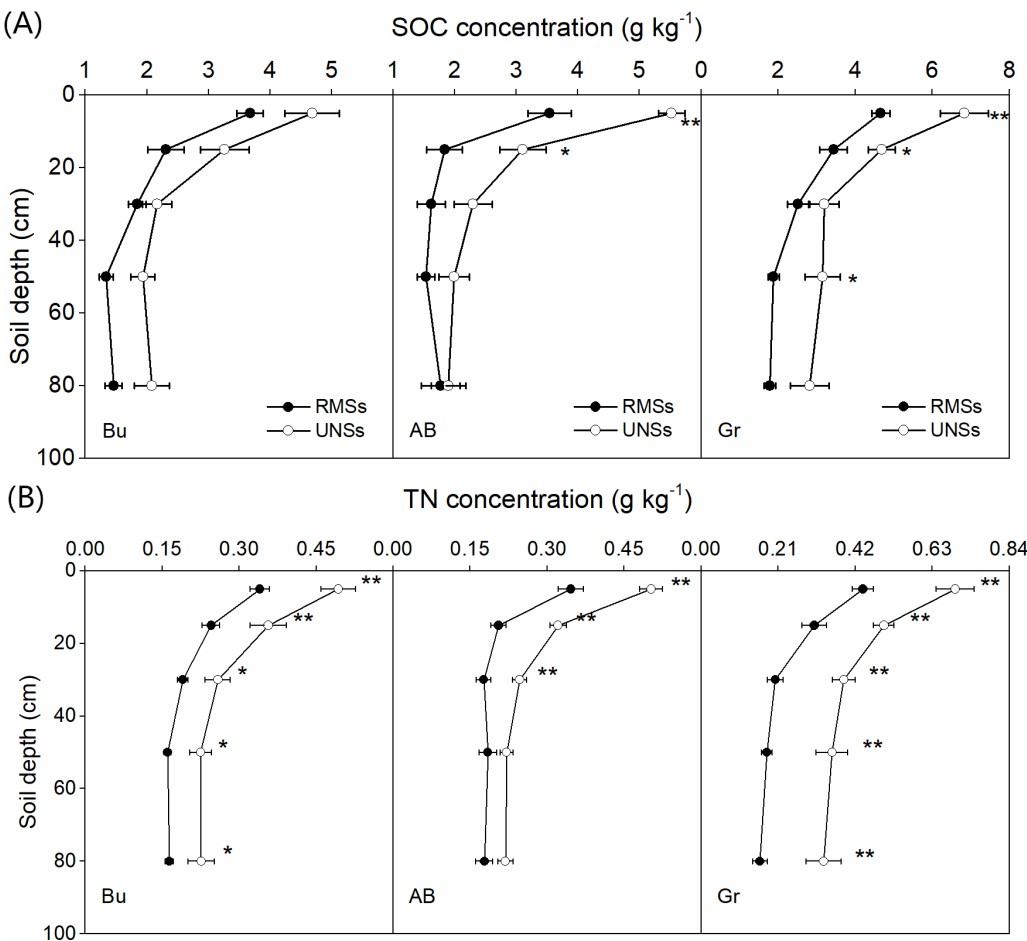

**Figure 3** Comparison of (A) SOC and (B) TN concentrations (g kg$^{-1}$) in reclaimed mine sites (RMSs) and undisturbed native sites (UNSs) under different vegetation types (Bu, bushes; AB, arbor-bush mixtures; and Gr, grasslands). ** and * represent significant differences between RMSs and UNSs at $p < 0.01$ and $p < 0.05$, respectively. Error bars show the standard deviations.

# DISCUSSION

Previous studies have demonstrated that surface mining and the initial stage of reclamation can result in significant losses of SOC and TN, mainly due to topsoil loss, mechanical mixing of the soil horizons, breakdown of soil aggregates, enhanced mineralization, erosion, leaching from the exposed soil surface, and the lack of new vegetation (*Shrestha & Lal, 2011*; *Zhou et al., 2017*; *Ahirwal & Maiti, 2018*; *Yuan et al., 2018*). *Li et al. (2014)* investigated the SOC and TN stocks in a newly constructed dump (with the same soil reconstruction practices as the northern dump but it was not revegetated) at the Heidaigou surface coal mine. In their study, the SOC stocks in the dump were 0.38 and 1.51 kg m$^{-2}$ in the 0–20 cm and 0–100 cm intervals, respectively, and the TN stock was 0.03 and 0.15 kg m$^{-2}$. These values represent losses of 69.8% and 66.7% for the SOC in the 0–20 cm and 0–100 cm intervals, respectively, and 76.9% and 65.9% for the TN compared with the mean values obtained for the UNSs in our study (Fig. 4). The large and relatively rapid declines

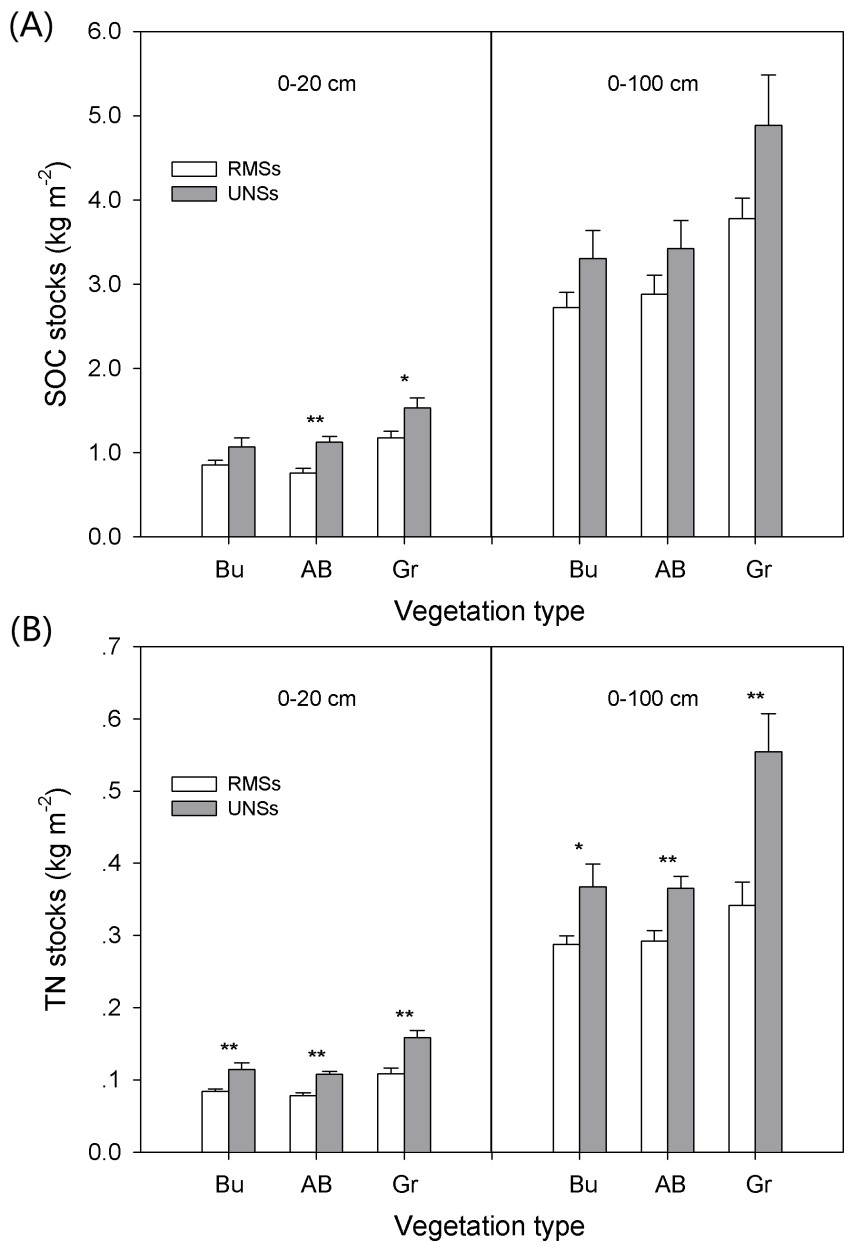

**Figure 4** **Comparison of (A) SOC and (B) TN stocks (kg m⁻²) in reclaimed mine sites (RMSs) and undisturbed native sites (UNSs) under different vegetation types (Bu, bushes, AB, arbor-bush mixtures, and Gr, grasslands).** ** and * represent significant differences between RMSs and UNSs at $p < 0.01$ and $p < 0.05$, respectively.

in the SOC and TN pools indicate a high potential for C and N accumulation in mine soils under appropriate reclamation practices (*Vindušková & Frouz, 2013*; *Mukhopadhyay & Masto, 2016*; *Ahirwal, Maiti & Reddy, 2017*). It has been reported that soils with a high potential for accumulating C and N are those with C and N contents below their carrying capacities, i.e., young soils or soils where C and N have been depleted because of

management practices, such as dramatically disturbed mine soils (*Glenn et al., 1993*; *Zhang et al., 2018a*; *Zhang et al., 2018b*).

Vegetation plays a major role in improving the properties of mine soils, where increased biomass production, root residues and exudates, and the greater activity of microbes and fauna following revegetation have positive effects on the accumulation of SOC and TN in RMSs (*Huang et al., 2016*; *Zhou et al., 2017*; *Yuan et al., 2018*). In agreement with previous studies (*Ahirwal, Maiti & Reddy, 2017*; *Ahirwal & Maiti, 2018*), we found that the increases in the SOC and TN stocks were more pronounced in the top layer (0–20 cm), where approximately one-third of the SOC and TN stocks were present. The SOC and TN stocks increased by 118.3% and 176%, respectively, in the 0–20 cm depth at the RMSs after 10–13 years of reclamation, compared with the stocks measured at the newly constructed dump (*Li et al., 2014*). The changes in the SOC and TN stocks agreed with the results of obtained by *Ahirwal, Maiti & Singh (2017)*, who reported that the SOC and TN stocks in the 0–30 cm depth of a reclaimed mine soil increased two times after 7–11 years of revegetation. These results also confirm the effects of revegetation on the accumulation of SOC and TN.

The SOC and TN dynamics as well as the potential of accumulating SOC and TN in mine soils are strongly related to revegetation patterns (*Chatterjee et al., 2009*; *Ganjegunte et al., 2009*; *Shrestha & Lal, 2010*; *Datta et al., 2015*). In the present study, the SOC and TN concentrations and stocks were significantly greater in the ARSs than the NRS (especially in the top 0–40 cm soil layer). *Liu, Lei & Gong (2019)* also found that the organic matter level in the 5–30 cm depth was significantly higher in ARSs (8.75%) than in an NRS (4.72%) in the Shendong mining area of China, which may have been due to the rapid growth of vegetation and biomass accumulation in the ARSs, thereby increasing the SOC and TN inputs from plant litter and root residues (*Liu, Lei & Gong, 2019*). Indeed, more plant litter (broken branches and fallen leaves) was found on the ground surface at the ARSs in our study (Fig. 5). Moreover, *Liu, Lei & Gong (2019)* reported that the microbial activity was higher in the RMSs, which was beneficial for the decomposition of plant residues, thereby leading to higher SOC and TN levels. However, the restoration of the soil quality in the disturbed mine land via natural succession is usually challenging because of its disordered strati-graphic sequence, server compaction, complicated surface materials, and degraded soil properties (*Zhou et al., 2017*; *Ahirwal & Maiti, 2018*; *Yuan et al., 2018*). *Bradshaw (1997)* estimated that it may take 50–100 years to restore the soil quality of a disturbed mine land to a similar state as the native soil by natural succession.

Among the artificial revegetation approaches, grasslands were more beneficial for the accumulation of SOC and TN than woodlands at the reclamation sites, especially in the top 40 cm of the soil. This result is consistent with those obtained by *Shrestha & Lal (2007)*, who showed that the SOC and TN contents of pasture lands were 99% and 98% higher than those of forested lands, respectively, at a 28-year-old reclaimed mine. These different capacities of soils for SOC and TN storage can probably be attributed to the changes in the amounts and forms of organic matter in the topsoil under different vegetation types (*Wei et al., 2009*), as well as the local climate conditions (*Mukhopadhyay & Masto, 2016*). In the woodlands, litterfall on the soil surface is the source of primary organic matter input in the topsoil, whereas the major organic matter input source is the decomposition of

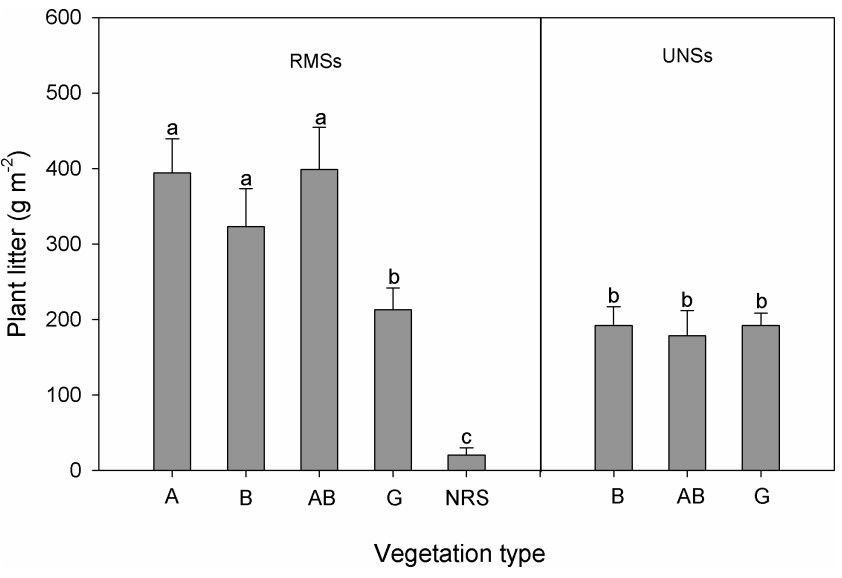

**Figure 5** **Plant litter in RMSs and UNSs under different vegetation types (Ar, arbors; Bu, bushes; AB, arbor-bush mixtures; Gr, grasslands; and NRS, natural recovery site).** Different lowercase letters represent significant differences between the vegetation types in RMSs and UNSs ($p < 0.05$). Error bars show the standard deviations.

belowground roots in grasslands (*Guo & Gifford, 2002*; *Wei et al., 2009*). This study area was arid with little precipitation, so the soil water supply was restricted to shallow soil horizons, which is more suitable for the growth and development of grasses. Grasses rapidly increase the root biomass and litter production, especially the growth of fine roots, which can fix large amounts of SOC and TN and transfer SOC and TN to the topsoil (*Jackson, Mooney & Schulze, 1997*). *Guo, Wang & Gifford (2007)* found that approximately 3.6 Mg C ha$^{-1}$ and 81.4 Mg N ha$^{-1}$ were contributed annually to the soil under a native pasture due to fine root mortality. Moreover, the reduced exchange of water and gases in grasslands because of the dense root network may reduce the turnover rate of SOC and TN (*Yakimenko, 1998*). By contrast, coarse roots represent most of the standing root crops in woodlands, which do not die and decompose for many years; thus, root production and turnover are of minor significance (*Guo, Wang & Gifford, 2007*). Although woodlands can produce more aboveground plant litter on the soil surface, the litter may prohibit precipitation from permeating into the mineral soil, which may facilitate the decomposition of organic matter and soil respiration but limit the plant uptake of precipitation (*He et al., 2008*). In addition, despite the comparable amounts of aboveground plant litter, arbor planting resulted in significantly lower SOC and TN values compared with those associated with shrubs, which was largely due to the microclimate. Compared with the exposed arbor lands, the high canopy density of the bush lands created a more humid environment, which is beneficial for the accumulation of SOC and TN following forestation (*Paul, Polglase & Richards, 2003*; *Wei et al., 2009*).

Overall, the SOC and TN concentrations in the deep soil intervals (below 40 cm) varied little among the revegetation patterns. Similarly, previous studies also indicated that the SOC and TN differences between vegetation types decreased with the soil depth (*Datta et al., 2015*), which may be attributed to the reduced residue inputs in the deep soil, especially those in the form of fine roots. *Zhang et al. (2011)* reported that the distribution of fine roots agreed with the distributions of SOC and TN. On the northern Loess Plateau, *Wei et al. (2009)* found that fine roots were mainly distributed in the 0–40 cm interval, and they accounted for 72%–94% of the fine roots in the 0–100 cm interval.

After reclamation for 10–13 years, the SOC and TN levels in the RMSs were still lower than those in the UNSs (only considering the shared vegetation types, i.e., Bu, AB, and Gr). The SOC stocks in the RMSs and UNSs only differed in the top 0–20 cm of the soil, but the differences were not significant when the entire soil profile (0–100 cm) was considered, and similar results were obtained by *Chatterjee et al. (2009)*. Based on the entire soil profile, the SOC concentrations in the RMSs were lower than those in the UNSs in the present study, i.e., by 24.8%–31%, and the TN concentrations were 27.8%–43.1% lower. Similarly, *Ahirwal, Maiti & Singh (2017)* reported that the SOC and TN levels determined at reclaimed sites after 11 years of reclamation were approximately 75% and 39% of those at the reference forest site, respectively. These results indicat that restoring the original SOC and TN levels after reclamation requires a significant period of time. According to *Huang et al. (2016)*, the SOC and TN levels in a revegetated site may require 23 to 25 years to reach the same levels as those in an undisturbed site. However, *Yuan et al. (2018)* suggested that the nutrient levels in revegetated sites could reach the same levels as those in undisturbed sites within about 10 year. According to *Ussiri & Lal (2005)*, mining and the associated disturbances disrupt the original SOC and TN equilibrium in the soil, while the accumulation of SOC and TN by appropriate reclamation and management practices promotes stabilization at a new near steady state equilibrium. However, the length of time required to reach the new equilibrium is uncertain. The SOC and TN accumulation rates in reclaimed mine soils depend on the microclimate, physicochemical and biological properties of the mine soils, vegetation type, prevailing management practices, and the time after vegetation establishment (*Ahirwal, Maiti & Reddy, 2017*; *Ahirwal & Maiti, 2018*). Moreover, the new equilibrium may eventually be similar, lower, or higher than the pre-mining equilibrium (*Ussiri & Lal, 2005*).

## CONCLUSIONS

Surface mining activities led to dramatic losses of SOC and TN. The results obtained in this study demonstrate that appropriate reclamation approaches such as revegetation could restore the SOC and TN pools in RMSs. However, restoring the original SOC and TN levels after reclamation required a significant amount of time. The effectiveness of restoration varied among the revegetation pattern. The NRS had lower SOC and TN concentrations and stocks compared with those in the ARSs, possibly because of the poor plant growth. Among the vegetation types in the ARSs, the potential for accumulating SOC and TN was higher in Gr than woodlands (Ar, Bu, and AB). A comparison of the different woodlands

types indicated that Bu had the largest accumulation potential, and Ar and AB had similar potentials. The results suggest that grasslands were more favorable for the accumulation of SOC and TN than woodlands, especially arbor lands, in this arid area.

The results of this study have important implications for land reclamation and ecological restoration in the mine areas of Loess Plateau. The planting of grasses such as *Leymus chinensis, Leymus secalinus, Medicago sativa, and Artemisia argyi*, should be encouraged to facilitate the reclamation of mine soils in this region.

## ACKNOWLEDGEMENTS

We thank the anonymous reviewers and the journal editors for providing constructive comments and suggestions on the manuscript.

### Funding

This study was financially supported by the Strategic Priority Research Program of Chinese Academy of Sciences (No. 40020100) and the National Natural Science Foundation of China (No. 41501236). The funders had no role in study design, data collection and analysis, decision to publish, or preparation of the manuscript.

### Grant Disclosures

The following grant information was disclosed by the authors:
Strategic Priority Research Program of Chinese Academy of Sciences: 40020100.
National Natural Science Foundation of China: 41501236.

### Competing Interests

The authors declare there are no competing interests.

### Author Contributions

- Ping Ping Zhang conceived and designed the experiments, performed the experiments, analyzed the data, prepared figures and/or tables, authored or reviewed drafts of the paper, and approved the final draft.
- Yan Le Zhang, Jun Chao Jia, Yong Xing Cui and Xia Wang performed the experiments, authored or reviewed drafts of the paper, and approved the final draft.
- Xing Chang Zhang and Yun Qiang Wang conceived and designed the experiments, authored or reviewed drafts of the paper, and approved the final draft.

### Field Study Permissions

The following information was supplied relating to field study approvals (i.e., approving body and any reference numbers):

Field experiments were approved by the land environmental protection office of Shenhua Group Zhungeer Energy Co., Ltd. (project number: KZCX2-XB3-13-02).

## Data Availability

    The raw data is available in the Supplemental Files.

## Supplemental Information

Supplemental information for this article can be found online at http://dx.doi.org/10.7717/
peerj.8563#supplemental-information.

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
