# Peer review of "Revegetation pattern affecting accumulation of organic carbon and total nitrogen in reclaimed mine soils"

_PeerJ, doi:10.7717/peerj.8563_

## Round 0.1 · original submission · Major Revisions

Dear Authors,

I have now received the reports from two reviewers who both agreed on the novelty and validity of your findings.

However, especially reviewer #2, highlighted several issues, mainly related to the structure of the manuscript and to the need for further information on the experimental design, that should be addressed before reconsidering it for the second round of revisions.

Sincerely

Marco Cavalli

·

Basic reporting

The novelty of the study was “to evaluate the effects of different revegetation patterns on the SOC and TN pools in reclaimed mine soils”. is very clear.
[1] LN 97-98: The mine encompasses a total area of 52.11 km2 , and the mine elevation is between 1025 m and 1302 m.  I suggest that you make it clear about mine elevation 1025 m to 1302m, it is from MSL or ground level or base level? Also explain, is ita plan area or undulating topography and lowest point is 1025m?
[2] LN 108-110: I suggest, you need to change/ rewrite LN 108-110 “ In the process of reclamation, at least 1 m of top subsoil (the deep loess parent materials stripped during mining) was applied on the top of the dump and appropriate leveling was conducted to create a flat surface”.  for example, you can add “ In the process of technical reclamation, (add technical), 1m top subsoil.. leveled appropriately .. (just change little bit this sentence).
[3] LN 146: I suggest for better clarity of the Line No 146 “ The bulk density was then used to convert SOC and TN values from g kg –1 to kg m –2” write in the formula form so that it could be easily understood.
[4] Add site photographs of reclaimed mine sites (RMSs), including artificially revegetated sites (ARSs ; arbors [Ar], bushes [Bu], arbor- bush mixtures [AB], and grasslands [Gr]); a natural recovery site (NRS), and undisturbed native sites (UNSs),  if available.

Experimental design

No comments

Validity of the findings

No comments

Reviewer 2 ·

Basic reporting

The Article needs a better background about the importance of SOC and TN in soils and climate change.

Experimental design

The experimental design could be better explained. Is not clear the the topographical positions were samples were taken

Validity of the findings

The results and the discussions are well done. In some parts they are mixed, but I think they could be rearranged. To improve the validity of the findings is important the authors explain better the experimental design.

Annotated reviews are not available for download in order to protect the identity of reviewers who chose to remain anonymous.

---

## Round 0.2 · accepted · Accept

Your manuscript has been revised again and I've received very positive feedback from the reviewers. The manuscript is now ready to be accepted.

Congratulations.
Marco Cavalli

·

Basic reporting

No MS is technically sound after incorporation of all suggestions/ corrections.

Experimental design

Satisfactory.

Validity of the findings

Good

Additional comments

MS may be accepted for publication.

Reviewer 2 ·

Basic reporting

No comment

Experimental design

No comment

Validity of the findings

No comment

Additional comments

From my part the article can be accepted